# Four-band non-Abelian topological insulator and its experimental realization

Tianshu Jiang [1,3], Qinghua Guo[1,3], Ruo-Yang Zhang [1], Zhao-Qing Zhang[1], Biao Yang [1,2✉] & C. T. Chan [1✉]

Very recently, increasing attention has been focused on non-Abelian topological charges, e.g., the quaternion group $Q_8$. Different from Abelian topological band insulators, these systems involve multiple entangled bulk bandgaps and support nontrivial edge states that manifest the non-Abelian topological features. Furthermore, a system with an even or odd number of bands will exhibit a significant difference in non-Abelian topological classification. To date, there has been scant research investigating even-band non-Abelian topological insulators. Here, we both theoretically explore and experimentally realize a four-band PT (inversion and time-reversal) symmetric system, where two new classes of topological charges as well as edge states are comprehensively studied. We illustrate their difference in the four-dimensional (4D) rotation sense on the stereographically projected Clifford tori. We show the evolution of the bulk topology by extending the 1D Hamiltonian onto a 2D plane and provide the accompanying edge state distributions following an analytical method. Our work presents an exhaustive study of four-band non-Abelian topological insulators and paves the way towards other even-band systems.

[1] Department of Physics and Institute for Advanced Study, The Hong Kong University of Science and Technology, Hong Kong, China. [2] College of Advanced Interdisciplinary Studies, National University of Defense Technology, 410073 Changsha, China. [3]These authors contributed equally: Tianshu Jiang, Qinghua Guo. ✉email: yangbiaocam@nudt.edu.cn; phchan@ust.hk

n mathematics, Abelian operators are commutative, meaning that the result of two successive operations does not depend on the order in which they are written. If we focus on a single bandgap, then topological physical systems[1–6] are usually classified by Abelian groups, with the prime example being the tenfold classification[7,8] of Hermitian topological insulators and superconductors. Once multiple bandgaps are collectively considered, their coupling introduces richer physics that can make the classification non-Abelian[9–13]. A classic example is the quaternion group $Q_8 = \{+1, \pm i, \pm j, \pm k, -1\}$ with $i^2 = j^2 = k^2 = ijk = -1$, which has been used to classify the topological line defects in biaxial nematic liquid crystals[14]. Very recently, non-Abelian groups have been used to describe the admissible nodal line configurations[12,15,16], Dirac/Weyl point braiding[13,17,18], and intriguing triple nodal points[19–21] in PT (inversion and time-reversal) symmetric systems. When more bands are involved, richer non-Abelian topological charges emerge[9]. Especially for systems with an even number of bands, several new classes of non-Abelian topological charges deserve special attention. A simple argument for this is that the even-dimensional special orthogonal groups, i.e., $SO(2N)$, with $N$ indicating a positive integer, contain inversion symmetry, i.e., $-I_{2N}$ (the negative $2N \times 2N$ identity matrix).

## Results

**Non-Abelian topological charges in four-band models.** Here, for simplicity, we focus on a four-band PT symmetric system. Choosing an appropriate basis, the Hamiltonian can take real forms, i.e., $H(k) = H^*(k)$. When simultaneously considering all three bandgaps between any two adjacent bands, the configuration space of the Hamiltonian is $M_4 = O(4)/\mathbb{Z}_2^4$, with $O(4)$ being the 4-dimensional (4D) orthogonal group. This implies that the eigenstate frame remains intact under $O(4)$ rotation, while $\mathbb{Z}_2^4$ indicates that each eigenstate has a gauge freedom of $\pm 1$. The quantized charges that describe the underlying topology are found to be the non-Abelian-based homotopy group[9] $\pi_1(M_4) = Q_{16} = U_{n_i \in \{0,1\}}\{\pm e_1^{n_1} e_2^{n_2} e_3^{n_3}\}$, where $e_1, e_2,$ and $e_3$ are the basis vectors of real Clifford algebra $C\ell_{0,3}$ satisfying the relation $\{e_i, e_j\} = -2\delta_{ij}$ (see Supplementary Note 1). There are 16 elements in the group and 10 conjugacy classes in total (see Table 1, as indicated by the curly braces). Group multiplication can be simply carried out using the above relation, i.e., $(e_1 e_2)(e_1 e_3) = -e_1 e_1 e_2 e_3 = e_2 e_3$. Although the labels with the

Clifford algebra basis (see the 1st column of Table 1) are convenient for group multiplication, decoding the underlying physical meaning is not straightforward. To relate the charges to rotations of the eigenstates, we rename all the charges one-to-one, as shown in the 2nd column of Table 1. For example, we will see that $\pm q_{12}$ indicate that both the 1st and 2nd bands acquire Zak phases of $\pi$ due to the rotation of their respective eigenvectors. Figure 1a shows the representative elements and their multiplication relations, and the corresponding full multiplications are listed in Supplementary Tables 1 and 2. One may also note that the paths (arrows) bridging two elements are not unique. This means that the non-Abelian topological phase transitions are multiple-path transitions, which is different from the single-path transitions in Abelian systems[22].

In the following, we study the topological properties of these charges. After topological band flattening, the mentioned PT symmetric four-band Hamiltonian can take the form of $H(k) = R(k)I_{1234}(k)R^T$, with $R(k) \in SO(4)$ being the 4D special orthogonal group, $k \in [-\pi, \pi]$ being the first Brillouin zone (FBZ) and $I_{1234} = diag(1, 2, 3, 4)$. The Hamiltonian has four real eigenvectors, as $H(k)|n\rangle = n|n\rangle$ with $n =$ 1, 2, 3, and 4. When $k$ runs across the FBZ ($k = -\pi \rightarrow \pi$), rotation matrix $R(k)$ continuously acts on eigenvector $|n\rangle$, and one finally obtains $+ or - |n\rangle$ corresponding to a Zak phase of 0 or $\pi$, respectively. Without loss of generality, we assume $R(k = -\pi) = I_4$. Because $det(R) = \lambda_1 \lambda_2 \lambda_3 \lambda_4 = 1$, with $\lambda_i$ being the four eigenvalues of $R(k)$, three exhaustive categories of possibilities at $k = \pi$ can be easily found (see see Table 1) : (1) all four $\lambda_i = 1$; (2) two $\lambda_i = 1$, with the other two $\lambda_i = -1$; and (3) all four $\lambda_i = -1$.

The first category corresponds to two conjugacy classes $\{+1\}$ and $\{-1\}$. Although they are indistinguishable from the Zak phase description, charge $+1$ indicates that the trajectories of the eigenstate frame are contractible, while charge $-1$ indicates a noncontractible loop. Usually, charge $-1$ indicates that the eigenstate frame rotates by $2\pi$ in a rotation plane (or topologically equivalent configurations)[9,22]. We will see their difference more explicitly by extending the 1D Hamiltonian onto a 2D plane (Fig. 1d, e). The second category consists of six conjugacy classes that can be distinguished using single-band Zak phase arguments regarding which two of the four bands have Zak phases of $\pi$. In the last category, all eigenstates flip their sign after $k$ runs across the 1D FBZ. This category originates from the inversion symmetry ($-I_4$) mentioned above. The two group elements (classes) also share the same Zak phase distribution and are indistinguishable from the conventional Abelian arguments. Their difference is reflected in the eigenstate rotation sense in four dimensions.

With setting $(k) = \exp(\phi \sum_{i<j=1:4} n_{ij} L_{ij})$, we obtain the explicit form of the flat-band Hamiltonian, where six skew-symmetric matrices $L_{ij}$ with entries $\left(L_{ij}\right)_{a,b=1:4} = -\delta_{ia}\delta_{jb} + \delta_{ib}\delta_{ja}$ span the basis of Lie algebra $\mathfrak{so}(4)$, $\phi(k)$ is the rotation angle and $n_{ij}(k)$ determines the rotation plane. For example, the Hamiltonian of charge $q_{12}$ can be given with $R(k) = \exp\left(\frac{k+\pi}{2} L_{12}\right)$, while that of charge $-1$ can be obtained with $R(k) = \exp\left[(k + \pi) L_{12}\right]$. Except for the charges $\pm q_{1234}$, the rest have counterparts in the three-band systems[22] studied previously. Thus, we mainly focus on the charges $\pm q_{1234}$, which are unique in the four-band models.

While the non-Abelian topological charges are defined on 1D periodic lattices, their topological characteristics would be more straightforward to visualize after we generalize the 1D Hamiltonians onto a 2D extended plane, where each non-Abelian topological charge characterizing the 1D loop is reflected by the specific configuration of band degeneracies encircled by the 1D loop in the 2D plane. After trigonometrically expanding the

---

**Table 1 Categories of non-Abelian topological charges in four-band models.**

| $Q_{16}$: Clifford-basis label | $Q_{16}$: Band index label | Eigenvalues: $(\lambda_1, \lambda_2, \lambda_3, \lambda_4)$ |
|---|---|---|
| $\{+1\}, \{-1\}$ | $\{+1\}, \{-1\}$ | $(1, 1, 1, 1)$ |
| $\{\pm e_1\}$ | $\{\pm q_{12}\}$ | $(-1, -1, 1, 1)$ |
| $\{\pm e_2\}$ | $\{\pm q_{13}\}$ | $(-1, 1, -1, 1)$ |
| $\{\pm e_3\}$ | $\{\pm q_{14}\}$ | $(-1, 1, 1, -1)$ |
| $\{\pm e_1 e_2\}$ | $\{\pm q_{23}\}$ | $(1, -1, -1, 1)$ |
| $\{\pm e_1 e_3\}$ | $\{\pm q_{24}\}$ | $(1, -1, 1, -1)$ |
| $\{\pm e_2 e_3\}$ | $\{\pm q_{34}\}$ | $(1, 1, -1, -1)$ |
| $\{+e_1 e_2 e_3\}, \{-e_1 e_2 e_3\}$ | $\{+q_{1234}\}, \{-q_{1234}\}$ | $(-1, -1, -1, -1)$ |

The three categories can be further decomposed into 10 conjugacy classes forming the generalized quaternion group $Q_{16}$. For the four-band system separated by three bandgaps, if we label each band with Zak phases of 0 or $\pi$, then there are $2^3 = 8$ possibilities, corresponding to the eight different eigenvalue sets. There are two classes that go beyond the Zak phase description[9].

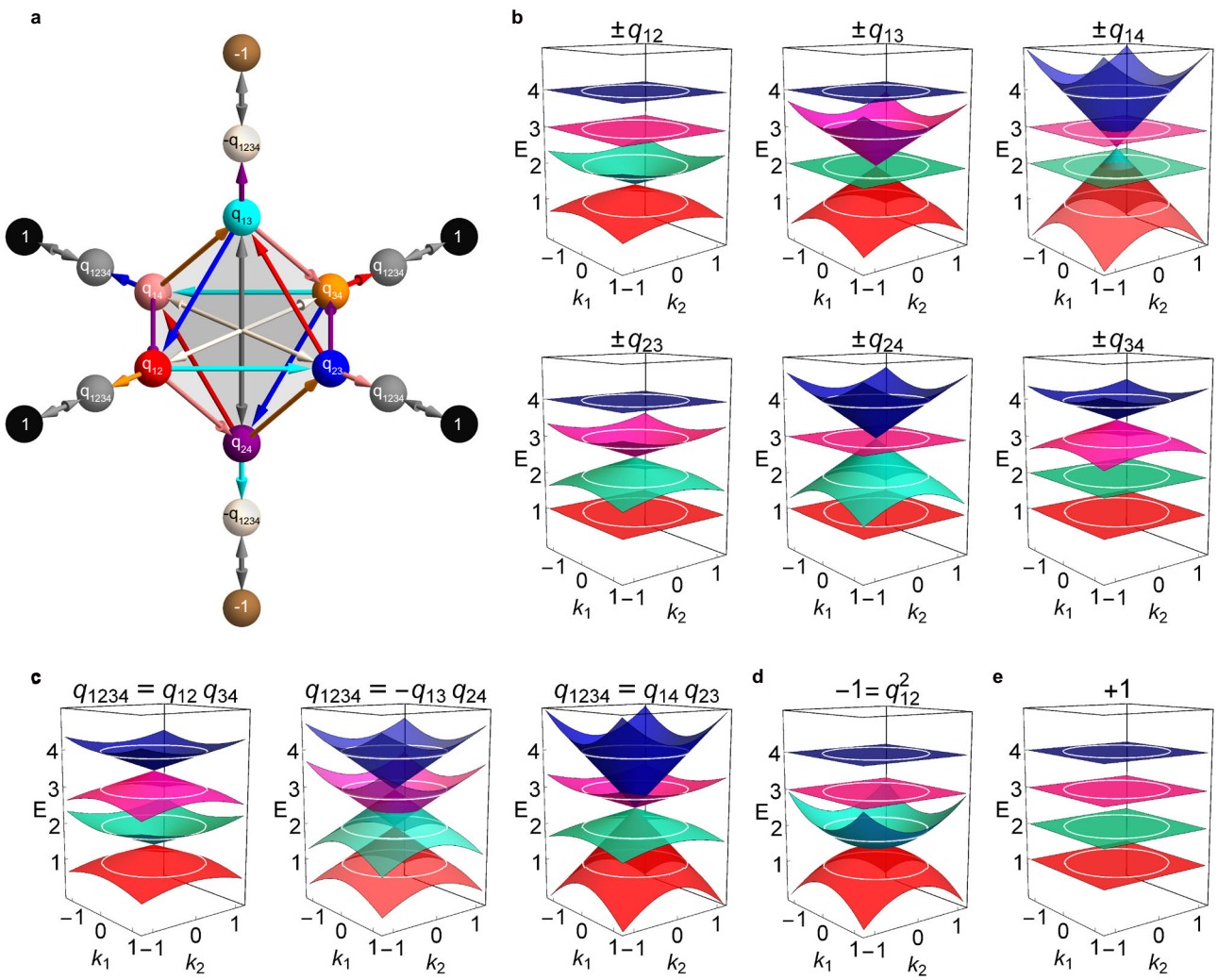

**Fig. 1 Non-Abelian topological charges in four-band models. a** Elements of the $Q_{16}$ group indicated by colored spheres sitting on an outstretched regular octahedron and their mutual multiplications represented by the corresponding colored arrows. For example, a red arrow $q_{12}$ brings a blue sphere $q_{23}$ to a cyan sphere $q_{13}$, indicating that $q_{23}q_{12} = q_{13}$. Full multiplication tables are provided in Supplementary Tables 1 and 2. **b–e** Extended 2D bands corresponding to three different categories of the non-Abelian topological charges: $\pm q_{mn}$, $q_{1234}$, and $\pm 1$. White circles indicate the corresponding 1D bands. The charge $q_{1234}$ can be decomposed in three ways in **c**, which are all topologically equivalent. For charge $-1$, we take $-1 = q_{12}^2$ as an example, and the other cases can be simply obtained by changing the linear Dirac cone degeneracies in panel **b** to quadratic degeneracies without any position shifting.

Hamiltonian $H(k)$, we make substitutions such as $\cos k \to \rho \cos k = k_1$ and $\sin k \to \rho \sin k = k_2$ and show the corresponding 2D bands in Fig. 1b–e. The original 1D Hamiltonian in $k$ space is a unit circle (white circles in Fig. 1b–e) in the 2D extended plane that encircles nonremovable degeneracies explicitly exhibiting the underlying topological obstacles. For the charge $+1$ (Fig. 1e), the topology is trivial, as there is no degeneracy enclosed by the white circles, while for charges $\pm q_{mn}$ (Fig. 1b) and $-1$ (Fig. 1d), the 1D unit circles enclose linear and quadratic degeneracies, respectively. These 2D degeneracies topologically contribute to edge/domain-wall states of the 1D systems; i.e., the linear/quadratic degeneracy implies one/two topologically protected edge states.

The charge $q_{1234}$ can be factorized as $q_{1234} = q_{12}q_{34}$, $q_{1234} = q_{14}q_{23}$, and $q_{1234} = -q_{13}q_{24}$ (the minus sign is induced by the odd permutation of subscripts). Note that the two factors in nodal links are commutative, i.e., $q_{12}q_{34} = q_{34}q_{12}$, which means that all nodes formed by more distant (i.e., sharing no common band) pairs of bands commute[9]. In Fig. 1c, we show the corresponding extended 2D band degeneracies of the three cases. They all belong to the same charge and can thus be continuously transformed into each other without closing the bandgap (see

below). The charge $-q_{1234}$ shares the same 2D band degeneracies with $q_{1234}$. Note that $\pm q_{1234}$ belong to two different conjugacy classes, which is one of the key points that fundamentally distinguishes them from the charges $\pm q_{mn}$. We will show their topological differences in the following section from the eigenstate rotation perspective.

We note that the nodal ring degeneracies in Fig. 1b ($\pm q_{14}$) and c ($-q_{13}q_{24}$) are accidental in the flat-band models, and each will be split into linear Dirac cones in more general situations (see below). Other triple degeneracies are similar to charges $\pm j$ in three-band models[22], where three bands are involved. The fourfold degeneracy in Fig. 1c ($q_{14}q_{23}$) is also admissible rather than stable here.

**Eigenstates on the three-sphere -$S^3$.** Here, we illustrate rotation configurations pertaining to different charges of the generalized quaternion group $Q_{16}$. The normalized eigenstates of $H(k)$ are all real and can be parametrized by Hopf coordinates $(\alpha, \eta, \beta)$ on the three-sphere - $S^3$. Their four components can be written as $(u = \cos\alpha\sin\eta, x = \sin\alpha\sin\eta, y = \cos\beta\cos\eta, z = \sin\beta\cos\eta)$, where

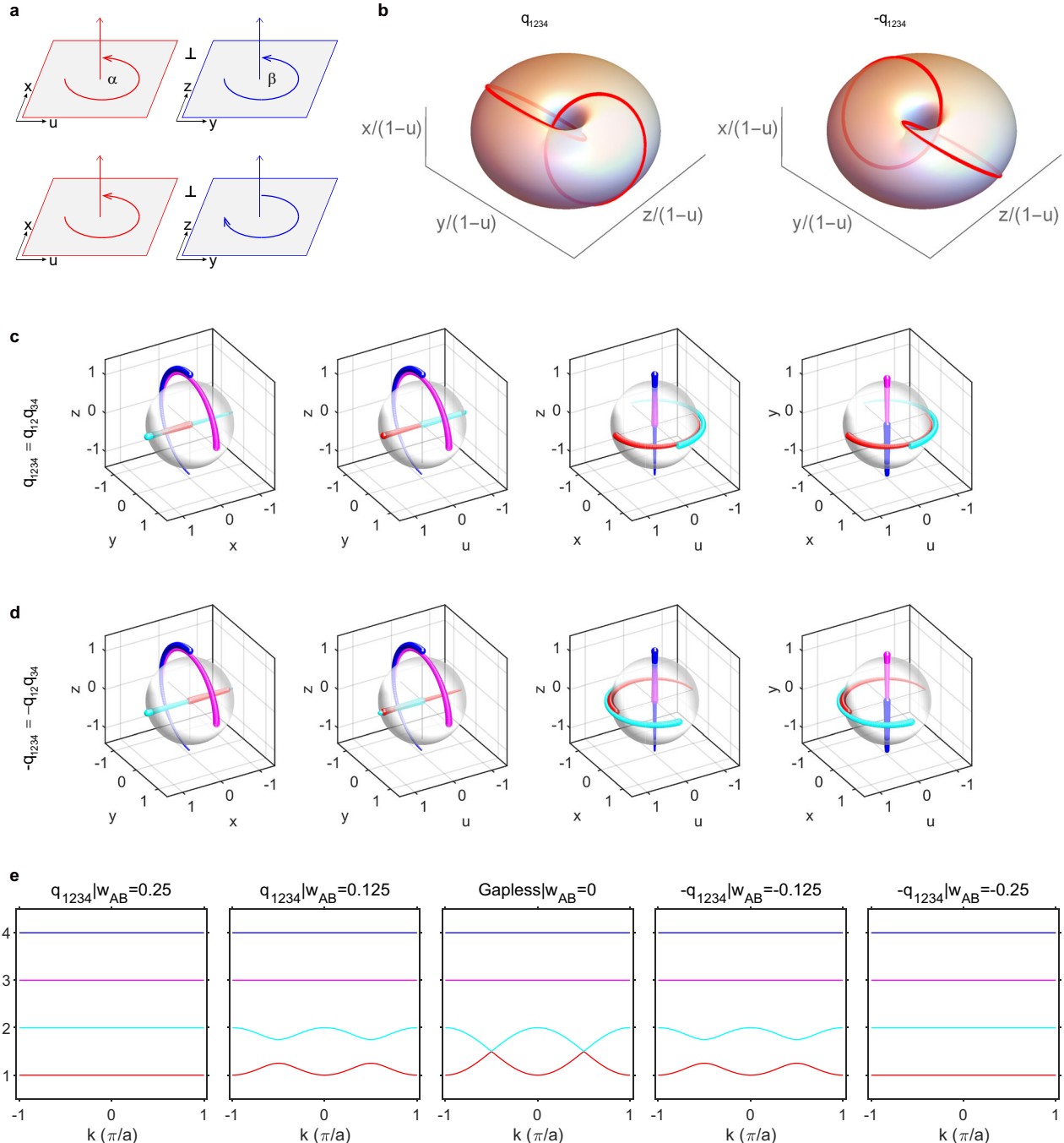

**Fig. 2 Non-Abelian topological charges $\pm q_{1234}$ specific to four-band models.** **a** Rotations in four dimensions. For each rotation $R$, there is at least one pair of two orthogonal rotation-invariant planes, e.g., $A = oux$ and $B = oyz$, that span the 4D space. For any $\vec{a} \in A$ and $\vec{b} \in B$, we have $\vec{a} \perp \vec{b}$, $R\vec{a} \in A$, and $R\vec{b} \in B$. We define the angle between $\vec{a}$ and $R\vec{a}$ ($\vec{b}$ and $R\vec{b}$) in the plane $A(B)$ as $\alpha(\beta)$. **b** 4D Clifford tori $(u, x, y, z)$ stereographically projected into $\mathbb{R}^3$ as the conventional tori $\left(\frac{x}{1-u}, \frac{y}{1-u}, \frac{z}{1-u}\right)$. The two linked circles represent the trajectory of one eigenstate (with applying all the 4D $D_2$ rotations), and the other three eigenstate trajectories overlap with this trajectory (see other cases in Supplementary Fig. 5). **c, d** Orthographic projections of the four eigenstate trajectories (shown in different colors) onto four 3D solid spheres, where one component is hidden by projection in each sphere. **e** Topological phase transition between charges $\pm q_{1234}$; other parameters are listed in Supplementary Table 4.

$\alpha$ and $\beta$ correspond to the two rotation angles in the two orthogonal invariant planes, as shown in Fig. 2a (also see Supplementary Note 2: Rotations in four dimensions[23–25]), while $\eta$ determines the proportions projected onto the two planes. When $\alpha \neq 0$ and $\beta = 0$ (or $\alpha = 0$ and $\beta \neq 0$), the rotations are called single rotations. For example, all ideal rotations $R(k)$ with $k = -\pi \rightarrow \pi$ enabling charge $q_{12}$ belong to the case with the settings $\eta = \frac{\pi}{2}$ and

$\alpha = \frac{k+\pi}{2}$, where "ideal" indicates the flat-band model given above. Note that all general models can be continuously transformed into the ideal flat-band model, and they are topologically equivalent. Other charges, including $\pm q_{mn}$ and $-1$, can be realized in a similar manner. Clearly, the eigenstates in one plane (i.e., $oyz$ plane when $\eta = \frac{\pi}{2}$) can be fixed for these cases, while they rotate on the other orthogonal plane (i.e., $oux$ plane). In

other words, the ideal rotations can be carried out in a 2D subspace. Notably, in contrast to on which plane the eigenstates rotate, the crucial property of these topological charges is that the eigenstate trajectories cannot contract to isolated points. The difference in charges $\pm q_{mn}$ is reflected by which two bands (the mth and nth) are noncontractible, while charge $-1$ requires that all four trajectories cannot contract simultaneously.

When both $\alpha \neq 0$ and $\beta \neq 0$, the rotations are dubbed double rotations (Fig. 2a), where there are two possibilities: rotating on the two planes in the same ($\alpha\beta>0$) or opposite ($\alpha\beta<0$) sense. The charges $\pm q_{1234}$ have to be realized with continuous double rotations, which means that $R(k)$ at each $k$ point is a double rotation. Interestingly, when $\eta = \frac{\pi}{4}$, the parametric set $(u, x, y, z)$ constructs a Clifford torus[26], which is the Cartesian product of two circles in $\mathbb{R}^4$ (e.g., $S_A^1 \in oux, S_B^1 \in oyz$ and $S_A^1 \times S_B^1 \in \mathbb{R}^4$). The Clifford torus can be stereographically projected[26] into $\mathbb{R}^3$ as a conventional torus, i.e., $\left(\frac{x}{1-u}, \frac{y}{1-u}, \frac{z}{1-u}\right)$, on which we can pictorially illustrate the difference between charges $\pm q_{1234}$ in the rotation sense of eigenstate trajectories, as shown in Fig. 2b. The two panels correspond to $\alpha = \beta = \frac{k+\pi}{2}$ (left, $q_{1234}$) and $\alpha = -\beta = \frac{k+\pi}{2}$ (right, $-q_{1234}$).

We further propose another orthographic projection method, which projects each 4D trajectory into 3D space from four orthogonal views. This is similar to the three-view drawing, which is the orthographic projection from 3D space to 2D plane. Taking the first panel of Fig. 2c as an example, we plot the trajectories in the $xyz$ subspace to obtain an orthographic projection from the view of the $u$ direction. Figure 2c, d correspond to $+q_{1234}$ and $-q_{1234}$, respectively, where eigenstate trajectories are mapped onto four solid spheres in $\mathbb{R}^3$. One can see that their main difference is that the rotation directions in the $oux$ plane are opposite. Orthographic projections for other charges are listed in Supplementary Figs. 1–4. In Fig. 2e, we show the topological phase transition between them, where there are inevitably two linear crossings between the first and second bands as system parameter $w_{AB}$ changes (without relying on a joint basepoint, as they belong to different classes).

**Zak phases and evolution of edge states.** After understanding the non-Abelian topological charges from the perspective of eigenstate frame rotations, we now show their relations to the Zak phases of each band as well as edge/domain-wall states. In a PT-symmetric system, the Zak phases of each band take a quantized value of 0 or $\pi$, and the values are shown in Table 1; i.e., $\lambda_i = -1$ indicates a Zak phase of $\pi$. We further refine the Zak phase of $\pi$ to be $\pm\pi$, where "$\pm$" is used to differentiate between charges $\pm q_{mn}$ (two elements in the same conjugacy class). All of the corresponding single-band Zak phases are exhaustively summarized in Fig. 3a. For charges $\pm q_{mn}$, two corresponding bands with noncontractible eigenstate trajectories carry Zak phases of $\pm\pi$, and the bandgap sandwiched by them supports edge states at hard boundaries of a finite lattice. We take the case of $\pm q_{12}$ as an example, as shown in Fig. 3b. The edge states of other $\pm q_{mn}$ charges are shown in Supplementary Fig. 6. We label charge $-1$ with $2\pi$, which indicates noncontractible $2\pi$ rotation here[22].

For charges $\pm q_{1234}$, two eigenstates rotate by $\pi$, while the other two rotate by $\pm\pi$ when $k = -\pi \to \pi$. As shown in Fig. 1c, there are three ways of factorization. We further schematically show them in Fig. 3c, where each double-headed arrow represents one factorization. The commutative property between two factor charges, i.e., $q_{12}q_{34} = q_{34}q_{12}$, is implied by the double-headed arrows. The fact that $q_{12}q_{34}$ (type-I), $-q_{13}q_{24}$ (type-II) and $q_{14}q_{23}$ (type-III) are the same element in the group can be visualized by constructing a transformation between them

without gap closing. The continuous transition between different factorizations can be explicitly parameterized. For example, from $q_{12}q_{34} \to -q_{13}q_{24}$, we have $H(k) = R_2 R_1 I_{1234} R_1^{-1} R_2^{-1}$, with $R_1(k) = \exp\left[(k+\pi)/2\left(\cos\theta_{I\to II}L_{12} - \sin\theta_{I\to II}L_{13}\right)\right]$ and $R_2(k) = \exp\left[(k+\pi)/2\left(\cos\theta_{I\to II}L_{34} + \sin\theta_{I\to II}L_{24}\right)\right]$, as shown in Fig. 3c (see the evolution of eigenstate trajectories in Supplementary Fig. 3). In other words, the pair of two orthogonal invariant planes rotates with $\theta_{I\to II}$. We further study the accompanying evolution of edge states at hard boundaries, as shown in Fig. 3d–f. The analytical results are $E^\pm = \frac{5}{2} \pm \frac{\sqrt{2}}{4}\sqrt{5 + 3\cos2\theta_{I\to II}}$, $E^\pm = \frac{5}{2} \pm \frac{1}{2}\cos\theta_{II\to III}$, and $E^\pm = \frac{5}{2} \pm \sin\theta_{III\to I}$. Detailed analytical methods are provided in Supplementary Note 3 and 4. There are a total of two edge states pumping between different bandgaps. Their field distributions are given in Supplementary Figs. 7–9. The existence of these edge modes can be heuristically inferred by examining the band degeneracies of the extended 2D model. In Fig. 3g–i, we show the radial cuts of their extended 2D bands, where one can easily find that each linear degeneracy point at $k_r = 0$ implies the position of each edge state in Fig. 3d–f, respectively. Note that in these flat-band cases, only the degeneracies at $k_r = 0$ imply topological edge states, while other degeneracies ($k_r \neq 0$) accidentally emerge from the 2D nodal rings (e.g., see Fig. 1c), which have no topological implication.

We also show the edge state evolution for charge $-1$ in Supplementary Figs. 10 and 11 (see the analytical solutions in Supplementary Note 4). Along the 12 edges of the charge $-1$ octahedron (Supplementary Fig. 10a), the evolution shows strong resemblance to the three-band models[22]. This occurs because only three bands participate in the edge state pumping. As such, all these transitions can be understood via the rotations of eigenstates in the subgroup $SO(3)$, while the fourth band is fully fixed and decoupled. One other important note is that there are 12 possible routes rather than 15 (naively from $C(6, 2) = 15$) because direct evolution between two orthogonal planes (or between the diagonal points linked by the dashed lines in Supplementary Fig. 10a, b), e.g., between $q_{12}^2$ and $q_{34}^2$, is impossible. We also find that the transition can take arbitrary routes on the 8 faces of the charge $-1$ octahedron (see an example in Supplementary Figs. 10d and 11). Supplementary Fig. 12 shows the evolution of 2D extended band degeneracies, which help us understand the pumping of edge states accordingly, e.g., the double quadratic or triple linear degeneracies at $k_r = 0$ predict the emergence of topological edge states[22].

**Observation of charges $\pm q_{1234}$ in a transmission line network.** To realize and characterize charges $\pm q_{1234}$, we designed a transmission line network[22,27,28] (see the sample photo in Supplementary Fig. 13) consisting of 11 unit cells. There are four meta-atoms A, B, C, and D in one unit cell. The real-space Hamiltonian reads (see details in Supplementary Note 3)

$$\mathcal{H} = \sum_n \left( \sum_{\substack{X=A,B,C,D \\ Y=A,B,C,D}} s_{XY} c_{X,n}^\dagger c_{Y,n} + \sum_{\substack{X=A,B,C,D \\ Y=A,B,C,D}} v_{XY} c_{X,n}^\dagger c_{Y,n+1} + \text{h.c.} \right)$$

where $c_{X,n}^\dagger$ and $c_{X,n}$ are creation and annihilation operators on the sublattice '$X/Y$' and site '$n$', respectively. To realize an explicitly real Hamiltonian in momentum space, we introduce imaginary hoppings[22]. More details on the experimental realization are provided in Methods section and Guo et al.[22].

The left two panels (BulkS) in Fig. 4a show the numerically calculated and experimentally measured energy bands. We plot the corresponding eigenstate trajectories of the four bands in Fig. 4c, d. In the experimental model, at each $k$, we can see that one rotation

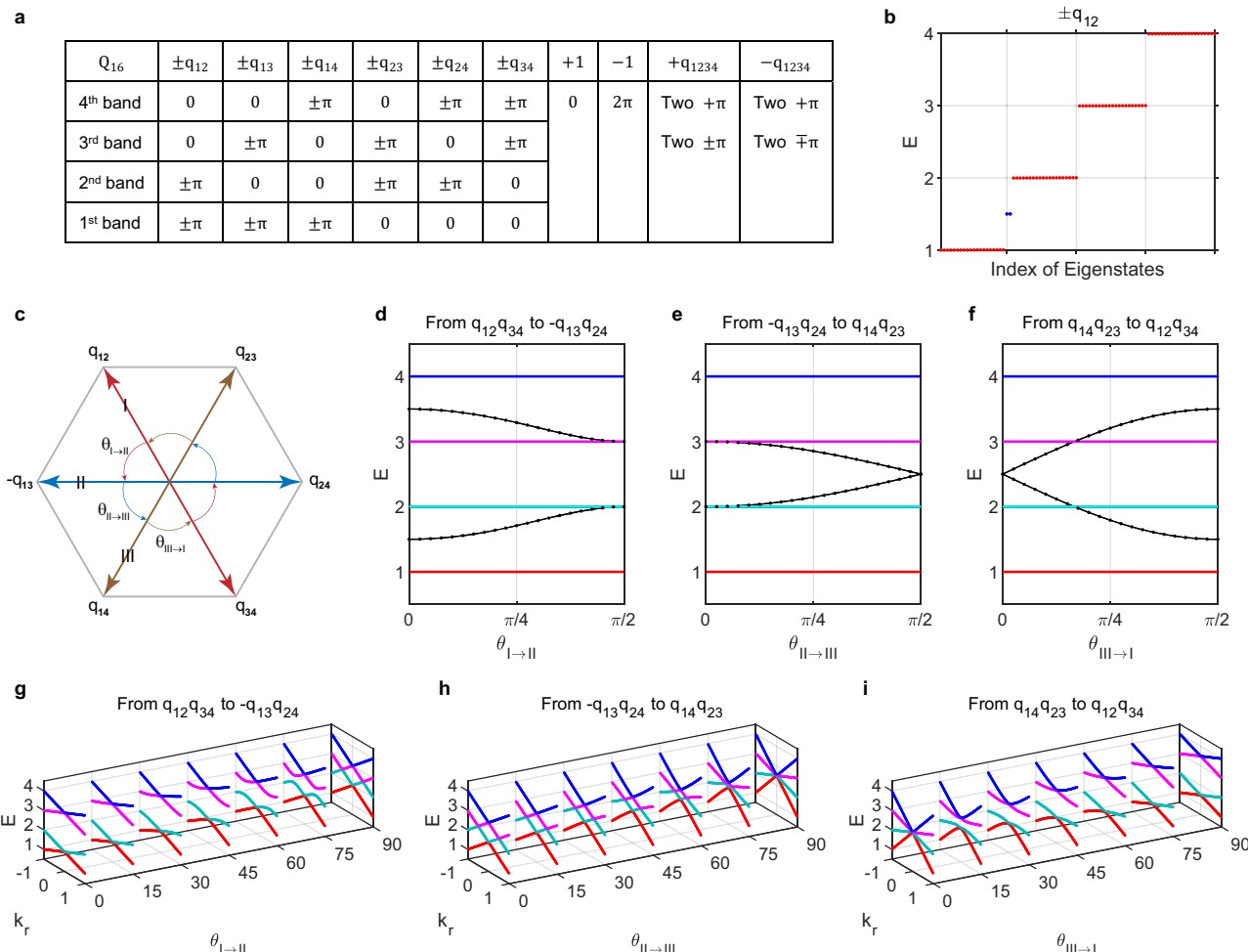

**Fig. 3 Zak phases and evolution of edge states. a** Zak phases for individual bands summarized for each non-Abelian topological charge. For charges $\pm q_{1234}$, two eigenvectors rotate by $\pi$, while the other two rotate by $+\pi$ or $-\pi$ depending on the factorizations (due to the handedness of the subspace). **b** Edge states of charges $\pm q_{12}$ occur at the bandgap sandwiched by the first and second bands. **c** Schematic view of charge $q_{1234}$ factorizations and their mutual continuous transitions. The double-headed arrows indicate that the paired two factors commute, i.e., $q_{12}q_{34} = q_{34}q_{12} = q_{1234}$. The directional arcs define the continuous transitions parametrized by $\theta_{m \to n}$, with $m$ and $n$ taking values of I, II, and III, corresponding to the factorizations of $q_{12}q_{34}$, $-q_{13}q_{24}$, and $q_{14}q_{23}$, respectively. **d-f** Evolution of edge states with varying parameter $\theta_{m \to n}$, where lines/dots indicate the numerical/analytical results. **g-i** Evolution of the extended 2D bands corresponding to (**d-f**), respectively. Note that we only plot the radial cuts $E(k_r)$, as the 2D bands for ideal flat-band models are isotropic in the $(k_1, k_2)$ plane.

plane is spanned by the eigenvectors of the first and second bands and the other rotation plane is spanned by those of the third and fourth bands. We can expect two topological edge states in total: one is located in the first bandgap (sandwiched by the first and second bands), while the other is located in the third bandgap (formed by the third and fourth bands). The rightmost panels (EdgeS) of Fig. 4a confirm our expectation. The detailed field distribution is provided in Supplementary Fig. 14b. The distribution of edge states can also be directly inferred from the 2D extended energy bands, as shown in Fig. 4b, where there is one linear Dirac cone between the first/third and second/fourth bands.

**Domain-wall states between charges $+q_{1234}$ and $-q_{1234}$.** If two samples with different non-Abelian topological charges meet at a domain wall[22], then some domain-wall states (DWSs) will emerge, and their existence can be predicted by defining a "domain-wall charge" $\Delta Q = Q_L/Q_R$. Here, $Q_L$ and $Q_R$ are the non-Abelian topological charges of the left and right samples, respectively. The quotient charge $\Delta Q$ is also an element of the non-Abelian group and governs the properties (including both

location and number) of the DWSs. We note that the appearance of the domain-wall charge $-1$ in the three-band system can only be well defined by assuming a joint $k$-space basepoint between the left and right samples[22]. Otherwise, one cannot distinguish two non-Abelian topological charges (e.g., $+i$ and $-i$) in the same conjugacy class of the three-band system, and thus, the domain-wall charge becomes ill-defined. In the four-band system, however, there exists a basepoint-free domain-wall charge taking a value of $-1$ between charges $+q_{1234}$ and $-q_{1234}$. This occurs because they belong to two different conjugacy classes.

In the experiment, we construct a domain wall (blue spheres in Fig. 4e) between charges $\pm q_{1234}$, as shown in Fig. 4e, where we flip the directions of imaginary hoppings between meta-atoms C and D, as denoted by the blue arrows, to realize the charge $+q_{1234}$ on the righthand side of the domain wall. Figure 4f shows the DWSs between them, where the left inset is the simulated energy levels and the right two insets indicate the measured spectra on the domain wall for two different excitation/probe locations accordingly. These results indicate that there are two nearly degenerate topological DWSs in the third bandgap. This is the

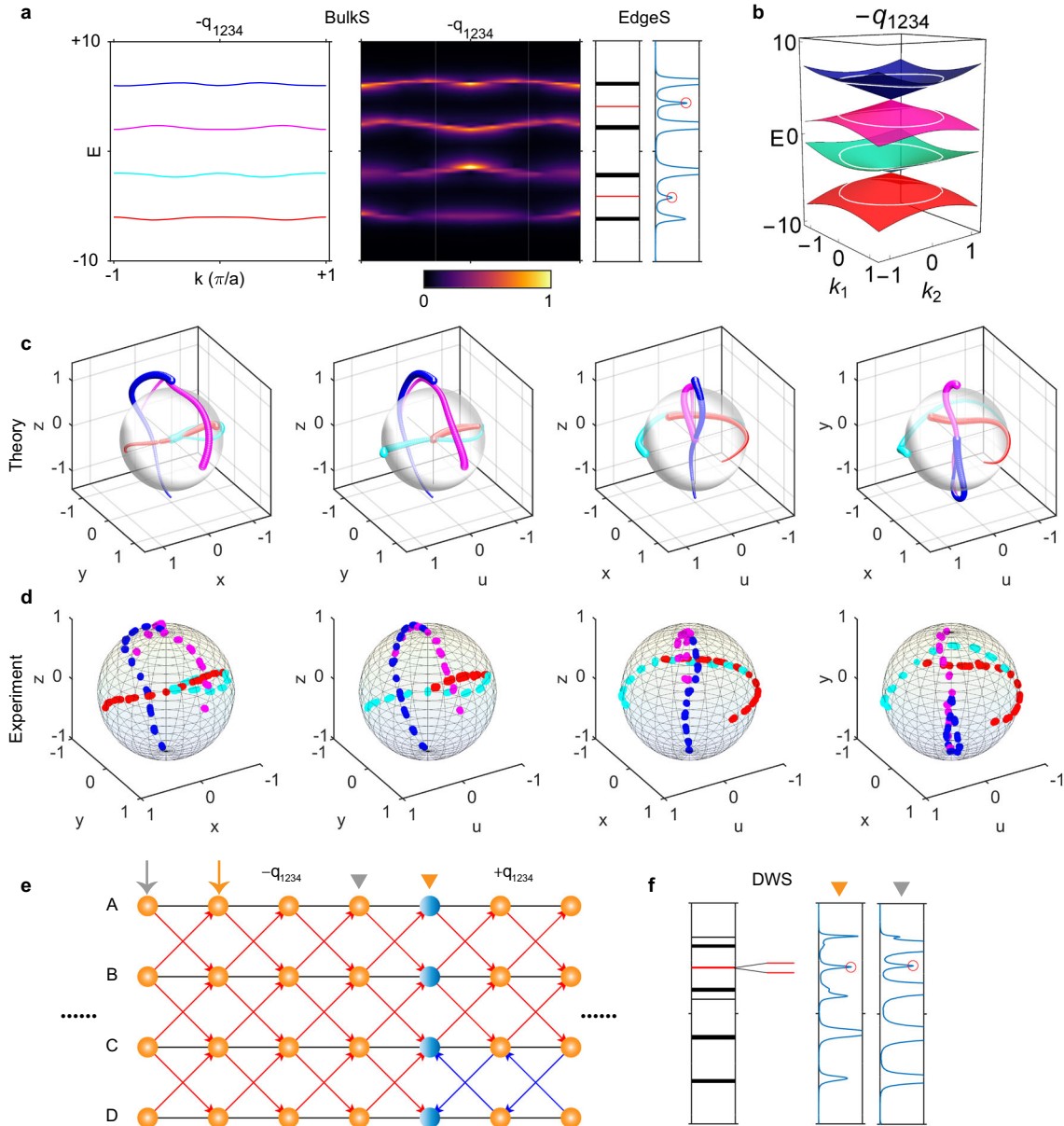

**Fig. 4 Experimental observation of the charge $-q_{1234}$ and edge/domain-wall states. a** The left panels show the numerically calculated and experimentally measured energy bands of bulk states (BulkS). The right panels display the energy spectra probed at the hard boundary, where the red lines and the peaks marked by red circles represent the simulation and experimental results of edge states (EdgeS), respectively. **b** Extended energy bands on a 2D plane; there is one linear Dirac cone between the first/third and second/fourth bands. White circles indicate the 1D energy bands. **c**, **d** Calculated/measured orthographic projections of eigenstate trajectories. The colors of trajectories correspond to different bands in **a**. The direction of decreasing linewidth indicates $k$ running from $-\pi$ to $\pi$. **e** Construction of the domain wall (marked by blue spheres) between charges $-q_{1234}$ and $+q_{1234}$. The gray and yellow arrows/triangles denote two different excitation/probe positions. **f** Calculated and measured energy spectra for the two different pairs of excitation and probe positions. The two domain-wall states are nearly degenerate. The detailed distribution of edge/domain-wall states is shown in Supplementary Figs. 14 and 15.

same as the hard boundary edge states of charge $-1$ and thus confirms our prediction. The detailed field distribution is provided in Supplementary Fig. 15.

**Observation of charges $\pm q_{14}$ in a transmission line network.** In addition, we experimentally studied charges $\pm q_{14}$, which are also interesting in the four-band models, as they exhibit three edge states in the three bandgaps. As shown in Fig. 5, from the bulk bands (Fig. 5a-BulkS), edge state distributions (Fig. 5a-EdgeS) and eigenstate trajectories (Fig. 5c, d), the numerical calculations correctly predict the experimental results. Different from Fig. 1b

$(\pm q_{14})$ of the flat-band model, the 2D extended energy bands in Fig. 5b are bridged by three linear Dirac cones. As mentioned above, each implies one edge state (per edge), as verified in Fig. 5a-EdgeS. For charges $\pm q_{14}$, there is no complete bandgap in the 2D extended bands. They can be regarded as the generalization of charges $\pm j$ in three-band models[9,22].

**Discussion**

Other general configurations of charges $\pm q_{1234}$ are shown in Supplementary Figs. 16 and 17, corresponding to the factorizations of $-q_{13}q_{24}$ and $-q_{14}q_{23}$, respectively. As mentioned above,

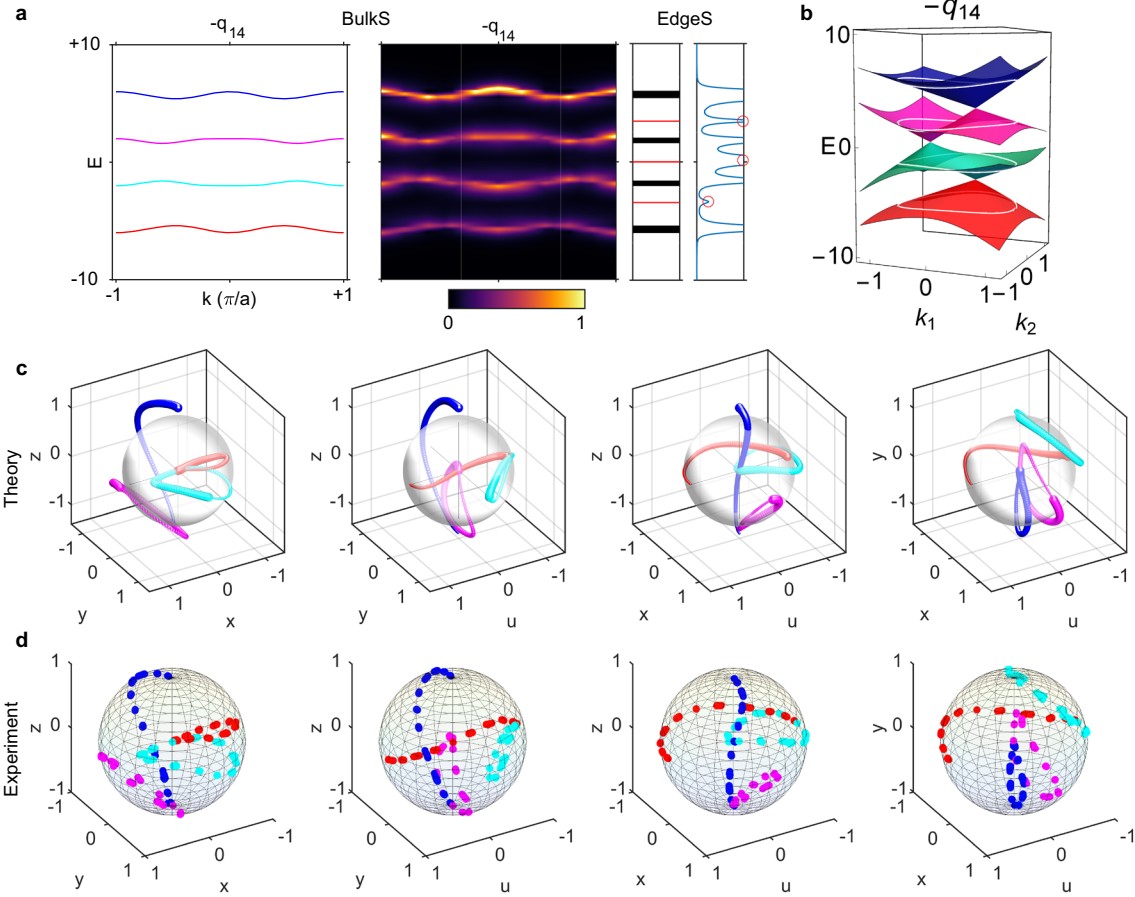

**Fig. 5 Experimental observation of the charge $-q_{14}$ and edge states. a** The left panels show the numerically calculated and experimentally measured energy bands of bulk states (BulkS). The right panels display the energy spectra probed at the hard boundary, where the red lines and the peaks marked by red circles represent the simulation and experimental results of edge states (EdgeS), respectively. **b** Extended energy bands on a 2D plane; there is one linear Dirac cone in each bandgap. White circles indicate the 1D energy bands. **c, d** Calculated/measured orthographic projections of eigenstate trajectories. The colors of trajectories correspond to different bands in **a**. The direction of decreasing linewidth indicates $k$ running from $-\pi$ to $\pi$.

the ring degeneracy formed by the second and third bands in Fig. 1c ($-q_{13}q_{24}$) splits into two Dirac cones, as shown in Supplementary Fig. 16c, which further implies two edge states (per edge) in Supplementary Fig. 16d. The general model of charge $-1$ in Supplementary Fig. 18 shows one triple linear degeneracy, which is similar to what we have observed in the three-band models, such as the edge state distributions[22].

In the classic context, the transmission line network is extremely versatile and can be used to realize various lattice models, including path-dependent annihilation of Dirac points in 2D, braiding of Weyl points in 3D[13], accompanying dynamics of wave-packet propagation, etc. In the quantum context, all of the single-particle topological phenomena can be well transferred. Furthermore, with introducing extra interaction and correlation physics, much more exotic topological braiding features are expected. Currently, the non-Abelian topological charges are limited to PT symmetry, it is very desirable to extend to other symmetry protections as well as non-Hermitian systems[29].

Our exhaustive study of all non-Abelian topological charges of PT symmetric four-band Hamiltonians will constructively stimulate related research on 2D twisted bilayer graphene[10,30,31]. The PT symmetric system also contributes to exotic fragile topological states[32] and even topological effective gravitational theory[33]. The studies can be easily transferred to other artificial platforms, including optical lattices[34], photonics[35–37], and phononics[38].

## Methods

**Experimental measurements**. There are four meta-atoms A, B, C, and D in one unit-cell. The hoppings between two meta-atoms are realized by connecting 2-m-long coaxial cables (model: RG58C/U). To achieve the complex hoppings, we create a hidden dimension by placing four nodes in each meta-atom so that four subspaces are allowed. Due to periodic connections in this hidden dimension, the four subspaces correspond to four pseudo angular momenta that are $\exp(i4\varphi_n) = 1$, with $\varphi_1 = 0$, $\varphi_2 = \frac{\pi}{2}$, $\varphi_3 = \pi$ and $\varphi_4 = -\pi/2$. Through the specific excitation from a 4-channel signal generator (Keysight M3201A), we carried out our experiments in the $\varphi_2 = \pi/2$ subspace. The amplitude and phase of voltage of each meta-atom are probed by an oscilloscope (Keysight DSOX2002A). After subsequent Fourier transformation, we obtain the energy bands and eigenstates in the momentum space. Supplementary Fig. 13 shows the specific transmission line network corresponding to charges $\pm q_{14}$.

## Data availability

The experimental data that support the findings of this study are available in DataSpace@HKUST with the identifier "https://doi.org/10.14711/dataset/VNMSFX"[39].

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

## Acknowledgements

This work is supported by the Hong Kong RGC (AoE/P-502/20, 16310420, and 16307821), the Hong Kong Scholars Program (XJ2019007), the KAUST CRG grant (KAUST20SC01) and the Croucher foundation (CAS20SC01).

## Author contributions

T.J., Q.G., B.Y. and C.T.C. conceived the idea; T.J. designed the transmission line network with input from Z.-Q.Z. and C.T.C.; T.J. and Q.G. carried out all measurements; T.J., Q.G., R.-Y.Z., Z.-Q.Z., B.Y. and C.T.C. developed and carried out the theoretical analysis; B.Y. and C.T.C. supervised the whole project. T.J., Q.G. and B.Y. wrote the manuscript and the Supplementary Information with input from all other authors.

## Competing interests

The authors declare no competing interests.
