## [Peer Review File · Nature Communications]

REVIEWERS' COMMENTS

Reviewer #1 (Remarks to the Author):

The authors present a rigorous theoretical and experimental study of a variety of non-Abelian topological charges in PT symmetric four-band Hamiltonians. The work is a nice collection of new results in this field, it is well presented and also contains convincing, although a bit limited, experimental results. I do believe the work is at the level of impact required for publication in Nature Communications, and I am happy to recommend it. I would ask the authors to consider expanding on the opportunities that non-Abelian topological charges offer in the context of classical and quantum technologies, at the moment the outlook section appears to be limited in scope.

Reviewer #2 (Remarks to the Author):

Jiang et al. theoretically and experimentally studied the four-band non-abelian band topology. Although it's just one typical system as theoretically studied in Ref.[9], their detailed discussion about the topological transition which is not detailed studied in Ref.[9] for the four-band system still could be very interesting. Besides, the extension of the concept of the Zak phase to the non-abelian Zak phase in a one-dimensional system could also be interesting. They theoretically proposed a tight-binding model and experimentally realized it to build a domain wall consisting of two one-dimensional systems with different non-abelian topological charges and detects the boundary states. Their studies are very solid. I would like to recommend it to be published in Nature communications.

Responses to Reviewer 1

The authors present a rigorous theoretical and experimental study of a variety of non-Abelian topological charges in PT symmetric four-band Hamiltonians. The work is a nice collection of new results in this field, it is well presented and also contains convincing, although a bit limited, experimental results. I do believe the work is at the level of impact required for publication in Nature Communications, and I am happy to recommend it. I would ask the authors to consider expanding on the opportunities that non-Abelian topological charges offer in the context of classical and quantum technologies, at the moment the outlook section appears to be limited in scope.

Response:

We thank the referee for his/her efforts for reviewing our work and the recommendation for publication. For the interesting suggestion that expanding the concept of non-Abelian charges in the contexts of classical and quantum technologies, we have added the following sentences in the discussion part of the revised text,

“In the classic context, the transmission line network is extremely versatile and can be used to realize various lattice models, including path-dependent annihilation of Dirac points in 2D, braiding of Weyl points in 3D, accompanying dynamics of wave-packet propagation, etc. In the quantum context, all of the single-particle topological phenomena can be well transferred. Furthermore, with introducing extra interaction and correlation physics, much more exotic topological braiding features are expected. Currently, the non-Abelian topological charges are limited to PT symmetry, it is very desirable to extend to other symmetry protections as well as non-Hermitian systems.”

Responses to Reviewer 2

Jiang et al. theoretically and experimentally studied the four-band non-abelian band topology. Although it's just one typical system as theoretically studied in Ref.[9], their detailed discussion about the topological transition which is not detailed studied in Ref.[9] for the four-band system still could be very interesting. Besides, the extension of the concept of the Zak phase to the non-abelian Zak phase in a one-dimensional system could also be interesting. They theoretically proposed a tight-binding model and experimentally realized it to build a domain wall consisting of two one-dimensional systems with different non-abelian topological charges and detects the boundary states. Their studies are very solid. I would like to recommend it to be published in Nature communications.

Response:

We thank the referee for his/her efforts for reviewing our work and the positive assessments on our work.